# A Dual Role of Heme Oxygenase-1 in Angiotensin II-Induced Abdominal Aortic Aneurysm in the Normolipidemic Mice

**DOI:** 10.3390/cells10010163

**Published:** 2021-01-15

**Authors:** Aleksandra Kopacz, Damian Klóska, Ewa Werner, Karolina Hajduk, Anna Grochot-Przęczek, Alicja Józkowicz, Aleksandra Piechota-Polańczyk

**Affiliations:** 1Department of Medical Biotechnology, Faculty of Biochemistry, Biophysics and Biotechnology, Jagiellonian University, 30-392 Krakow, Poland; aleksandra.kopacz@doctoral.uj.edu.pl (A.K.); damian.kloska@uj.edu.pl (D.K.); ewa.werner@uj.edu.pl (E.W.); karolina.hajduk@uj.edu.pl (K.H.); anna.grochot-przeczek@uj.edu.pl (A.G.-P.); alicja.jozkowicz@uj.edu.pl (A.J.); 2Department of Animal Reproduction, Anatomy and Genomic, Faculty of Animal Science, University of Agriculture, 30-059 Krakow, Poland

**Keywords:** heme oxygenase 1, HO-1, abdominal aortic aneurysm, AAA, cardiovascular system, angiotensin II

## Abstract

Abdominal aortic aneurysm (AAA) bears a high risk of rupture and sudden death of the patient. The pathogenic mechanisms of AAA remain elusive, and surgical intervention represents the only treatment option. Heme oxygenase-1 (HO-1), a heme degrading enzyme, is induced in AAA, both in mice and humans. HO-1 was reported to mitigate AAA development in an angiotensin II (AngII)-induced model of AAA in hyperlipidemic ApoE^-/-^ mice. Since the role of hyperlipidaemia in the pathogenesis of AAA remains controversial, we aimed to evaluate the significance of HO-1 in the development and progression of AAA in normolipidemic animals. The experiments were performed in HO-1-deficient mice and their wild-type counterparts. We demonstrated in non-hypercholesterolemic mice that the high-dose of AngII leads to the efficient formation of AAA, which is attenuated by HO-1 deficiency. Yet, if formed, they are significantly more prone to rupture upon HO-1 shortage. Differential susceptibility to AAA formation does not rely on enhanced inflammatory response or oxidative stress. AAA-resistant mice are characterized by an increase in regulators of aortic remodeling and angiotensin receptor-2 expression, significant medial thickening, and delayed blood pressure elevation in response to AngII. To conclude, we unveil a dual role of HO-1 deficiency in AAA in normolipidemic mice, where it protects against AAA development, but exacerbates the state of formed AAA.

## 1. Introduction

Abdominal aortic aneurysm (AAA) is one of the most prevalent age-related cardiovascular disorders. It is a local at least 50% dilatation of the abdominal aorta occurring in the infrarenal area. Due to its asymptomatic development and progression, AAA is often overlooked and diagnosed late, bearing a high risk of rupture and sudden death [1]. Besides, upon diagnosis, the employed static imaging techniques give only partial information about rupture risk [2]. Therefore, it is essential to understand the mechanisms and factors triggering this aortic malformation and regulating the aortic integrity to provide adequate prevention or surveillance for patients.

Formation of AAA is accompanied by extensive activation of proteolysis, high inflammation, oxidative stress, which lead to smooth muscle cells death and aortic wall rearrangement. However, the detailed activation pathways are still not well elucidated [1]. One of the features in many human AAA is the thrombus formation, which can be either outside or within the aortic wall and is penetrated by blood components. There, trapped and damaged erythrocytes release prooxidant heme, which induces and activates heme oxygenase-1 [3]. 

Heme oxygenase-1 (HO-1, encoded by *HMOX1* gene) is an inducible rate-limiting enzyme, degrading heme into ferrous ions, carbon monoxide, and biliverdin. HO-1 activity can be altered by porphyrins, and the species of cation of metal linked to porphyrin rings determines the direction of the change. Those modulators of HO-1 activity, together with knockout models, were used throughout years to demonstrate high antioxidant, anti-inflammatory, and antiapoptotic properties of its metabolites in numerous models [4]. These qualities may play a pivotal role in the development of AAA (further reviewed in [3]). In addition, *HMOX1* gene promoter polymorphism affects the expression of HO-1 in humans, suggesting an additional indication for the clinical relevance of studies on the role of HO-1 in AAA pathophysiology [5].

Up to date, the role of HO-1 in AAA has not been fully elucidated as the available reports draw contrasting conclusions. Some of them indicate for detrimental role of HO-1. It is upregulated in human AAA [6], and we recently showed that its level positively correlates with the aortic diameter [7]. In accordance, the upregulation of HO-1 was suggested to be associated with inflammation, instead of the protective effects in AAA patients, and may represent a marker of disease severity [8]. On the other hand, Austrian patients with AAA are characterized by promoter polymorphic variants which lead to overall lower expression of HO-1 in the body [9]. Still, such a correlation was not found among Croatian patients [10]. The murine models of AAA imply a protective role of HO-1 [11,12]. However, these results were obtained in models characterized by a high incidence of AAA: infusion of angiotensin II to hypercholesterolemic ApoE^-/-^ mice and porcine elastase model [13]. Moreover, over the last two decades, a debate has arisen on the significance of hyperlipidemia in AAA development [14,15,16]. Therefore, in our study, we aimed to verify the impact of HO-1 deficiency on AAA development in non-hypercholesterolemic mice. We found that lack of HO-1 protects against the development of AAA in non-hyperlipidemic mice. However, the aneurysm may still form, and then they are significantly more prone to rupture. It is associated with delayed blood pressure elevation, changes in the expression of collagen, metalloproteinases, and its inhibitors.

## 2. Materials and Methods

### 2.1. Animals

Experiments were performed on HO-1^−/−^ mice and HO-1^+/+^ littermates both on mixed C57/Bl6 × FVB background. Six-month-old male mice of verified genotype were used (4–6 animals per group at each time point). The genotype was verified prior to the experiment by DNA analysis. The animals were maintained under specific pathogen-free conditions in the individually ventilated cages (14/10 h light/dark cycle at a temperature of 22 ± 2 °C) and were provided with a 25% fat diet and water ad libitum. All experimental procedures were approved by the Second Local Ethics Committee for Animal Experiments in Krakow (No. 74/2016 and 110/2016) and performed in accordance with the guidelines from Directive 2010/63/EU of the European Parliament on the protection of animals used for scientific purposes.

### 2.2. Development of Abdominal Aortic Aneurysm

Mice were infused via osmotic minipumps (Alzet 2004, Alzet Osmotic Pumps, Cupertino, CA, USA) with angiotensin II (1000 ng/kg/min or 2500 ng/kg/min in saline; Sigma-Aldrich, St. Louis, MO, USA) or saline (sham group) for 28 days. Osmotic pumps were placed subcutaneously under Avetin (TBE) anaesthesia (250 mg/kg b.w.; 1.5% solution, 0.5 mL) as described previously [17]. The following parameters were monitored every seven days: body weight, blood pressure (BP), aortic diameter, and appearance by ultrasonography (USG) (Appendix A). BP change after osmotic pump placement was additionally measured on day 2. The mice were sacrificed on day 14 or 28 by overdosing of carbon dioxide; blood and aortas were collected for further biochemical analyses. The aortas were cleaned from the adjacent connective tissue and fat and photographed on a scale ruler. They were further preserved in OCT freezing medium for histological staining or in RNAlater (Sigma-Aldrich, St. Louis, MO, USA) for gene expression analysis. In this study, all the analyses were performed on the abdominal part of the aorta.

### 2.3. Blood Pressure Monitoring

All mice underwent non-invasive blood pressure measurement by tail-cuff plethysmography (BP-2000 series II; Blood Pressure Analysis Systems, Visitech Systems, Napa, PI, USA) which was preceded by a period of adaptation. The measurements were performed by one person to decrease any bias. The results were presented as a percentage change compared to day 0 (after adaptation time and before the administration of any drugs). This way of data presentation was chosen to plot the impact of administered compounds better and to minimize the influence of phenotypic and age-related changes between littermates. Importantly, no difference in mean blood pressure between genotypes was observed at day 0.

### 2.4. In Vivo Ultrasound Imaging of Abdominal Aorta and Analysis

Changes in the abdominal aorta diameter were monitored using the high-frequency ultrasound imaging system (Vevo 2100, FUJIFILM VisualSonics, Amsterdam, The Netherlands). Two-dimensional (B-mode) imaging using a 22–55 MHz linear-array transducer (MS550D) synchronized to the electrocardiographic signal was done. The animals were placed in a supine position on a heated table under inhalation anaesthesia with isoflurane (1.5–2%). The abdominal cavity was shaved, and a pre-warmed ultrasound gel was applied to the area of interest. Longitudinal images of the suprarenal and infrarenal aorta and transverse images at the level of the abdominal aorta between the diaphragm and the outlet of the left renal artery were acquired in the B-mode and M-mode to assess the maximum cross-sectional diameter (during diastole) and aortic area in real-time for each mouse at each time point. The detailed analysis scheme is presented in [18].

### 2.5. Blood Cell Count

Right after euthanasia, approximately 1 mL of blood was collected from the retroorbital sinus to a tube coated with EDTA. Blood cell count was analyzed using ABC Vet Hematology Analyzer (Horiba, Kyoto, Japan).

### 2.6. Total RNA Isolation, Reverse Transcription and Quantitative PCR

RNA from 3 mm fragment of abdominal aortic tissue was extracted with the RNeasy Mini Kit (Qiagen, Duesseldorf, Germany) according to the manufacturer’s instructions. cDNA was synthesized using a High-Capacity cDNA Reverse Transcription Kit (Thermo Fisher Scientific, Waltham, MA, USA). RT-qPCR was conducted on Step-One Plus Real-Time PCR Systems using a Power SYBR^®^ Green PCR Master Mix according to the manufacturer’s instructions (Thermo Fisher Scientific, Waltham, MA, USA). Primer sequences are gathered in Appendix A. Eukaryotic mouse translation elongation factor 2 (*eEF2*) was used as a reference gene. Relative gene expression was calculated using the ΔΔCt method.

### 2.7. Histological and Immunofluorescent Stainings

Haematoxylin and eosin stainings were performed on 9 µm tissue slices using the method available on the IHCworld protocols website. Trichrome staining was done on 9 µm tissue slices using the Trichrome Stain (Masson) Kit (Sigma-Aldrich, St. Louis, MO, USA). Samples were analyzed under a light microscope (Nikon; Tokyo; Japan) with NIS elements BR software (Canon, Tokyo, Japan) at magnifications of 100× and 400×.

Immunofluorescent stainings were done in frozen 10 µm (collagens I and III, osteopontin, HO-1), or 30 µm (immune cells) specimens of the abdominal aorta. Samples were fixed with methanol or ice-cold acetone and blocked in 10% donkey serum with 0.05% Tween-20 for 1 h at room temperature (RT). After washing in PBS, samples were incubated overnight (4 °C) with antibodies: anti-HO-1 (Enzo, New York, NY, USA), anti-collagen I or collagen III (Abcam, Cambridge, UK), anti-osteopontin (Proteintech, Rosemont, IL, USA). On the next day, samples were washed and incubated with secondary donkey antibodies conjugated with either Alexa Fluor 488 or Alexa Fluor 568 (Life Technologies, Carlsbad, CA, USA) for 1 h at RT. Nuclei were counterstained with DAPI (Sigma-Aldrich, St. Louis, MO, USA). For immune cell detection, PE-conjugated antibodies were used: anti-CD45 (BD, clone 30-F11), anti-CD86 (BD, clone GL1), anti-CD163 (BD, clone TNKUPJ, NJ, USA). Samples were analyzed under a meta laser scanning confocal microscope (LSM-880, Carl Zeiss, Oberkochen, Germany) and analyzed using ImageJ software (Wayne-Rasband (NIH)).

### 2.8. Heme Level Measurement

Intracellular heme concentration was checked using a fluorescence-based assay. First, 3 mm part of abdominal aorta was rubbed and resuspended in 2 M oxalic acid (Sigma-Aldrich, St. Louis, MO, USA). Half of the sample volume was heated at 95 °C for 30 min leading to iron removal from heme. Fluorescence (excitation 400 nm– emission 662 nm) of the resultant protoporphyrin was assessed by Tecan Spectra II Microplate Reader (Tecan, Mannedorf, Switzerland). The endogenous protoporphyrin content (measured in unheated sample in oxalic acid) was subtracted. Data were normalized to total protein concentration in each sample.

### 2.9. In Situ Gelatin Zymography (ISZ)

In situ gelatinolytic activity in the media of frozen abdominal aorta was performed as described previously [18]. Slides were incubated with 20 µg/mL solution of fluorescein-conjugated, dye-quenched gelatin from pig skin (DQ™-gelatin, Thermo Fisher, Waltham, MA, USA) prepared in gelatinase reaction buffer (150 mM NaCl, 5 mM CaCl_2_, 0.2 mM NaN_3_, 50 mM Tris-HCl, pH 7.6) at 37 °C in a dark wet chamber for 2 h. Next, slides were washed in MilliQ water, fixed for 1 min in ice-cold acetone, and blocked in the wet, dark chamber with 10% GS with 0.05% Tween-20 for 1 h at RT. Further, scraps were overlaid with MMP-2 and MMP-9 antibodies (Cell Signaling Technology, Danvers, MA, USA) in 1% GS with 0.05% Tween-20 and left overnight at 4 °C in a dark wet chamber. On the next day, samples were washed and incubated with anti-rabbit antibodies conjugated with Alexa Fluor 568 (Life Technologies, Carlsbad, CA, USA). Nuclei were counterstained with DAPI (Sigma-Aldrich, St. Louis, MO, USA). Samples were mounted in DAKO mounting medium and analyzed under a meta-scanning confocal microscope LSM 880 (Carl Zeiss, Oberkochen, Germany) at magnification 400× and analyzed using ImageJ software.

### 2.10. In Vitro Assays

Human aortic smooth muscle cells (HASMC) were grown in Medium 231 (Gibco, Thermo Fisher Scientific, Waltham, MA, USA) supplemented with 10% FBS and 1% penicillin streptomycin. Cells were cultured at 37 °C in a humidified incubator in 5% CO_2_ atmosphere. Transfections of HASMC were performed using 20 nM siRNA targeted against human *HMOX1* (s11056) or scrambled siRNA (Life Technologies, Carlsbad, CA, USA) using Lipofectamine™ RNAiMAX Transfection Reagent (Life Technologies, Carlsbad, CA, USA) in Opti-MEM I Reduced Serum medium (Life Technologies, Carlsbad, CA, USA). At 48 h after transfection, the cells were incubated for 24 h with 50 nM AngII (Sigma-Aldrich, St. Louis, MO, USA). For proliferation assessment, cells were washed with PBS, fixed with ice-cold methanol, blocked in 3% BSA, and incubated with anti-PCNA antibody (Thermo Scientific, Waltham, MA, USA) at 4 °C overnight. Next day, cells were incubated with secondary antibody conjugated with Alexa Fluor 568 (Life Technologies, Carlsbad, CA, USA) and Hoechst 33342 to stain the nuclei.

### 2.11. Statistical Analysis

Data are presented as mean ± SEM. Two-way ANOVA, followed by Tukey’s post hoc test, was used for comparison of more than two groups. Fisher’s exact test was used to calculate the frequency of aneurysm appearance and rupture. Mann-Whitney test was used for comparison of 2 groups. Grubbs’ test was used to detect statistically significant outliers (*p* < 0.05), which were not included in the statistical analysis of the results (GraphPad Prism 8 software). *p* < 0.05 was accepted as statistically significant.

## 3. Results

### 3.1. Heme Oxygenase 1 Is Upregulated During the Angiotensin II-Induced Formation of AAA in Non-Hypercholesterolemic Mice

The commonly used dose of angiotensin II (AngII) in AAA-related studies is 1000 ng/kg/min; however, such a quantity leads to inefficient AAA formation in wild-type animals. Thus, in our study, we used the standard (1000 ng/kg/min; A1000) and high dose (2500 ng/min/kg; A2500) of AngII. Mice were kept on a fat-enriched diet, containing 25% saturated fat. Such a regime did not alter the plasma cholesterol and triglycerides level (Figure 1A,B), implying the non-hyperlipidemic state of mice. The infusion of a high dose of AngII led to the formation of AAA in around 80% of animals. The frequency of appearance was significantly greater than in the case of the standard dose (Fisher’s exact test, *p* < 0.05), which led to the formation of AAA in 30–40% of mice, as reported by several groups, including us [18] (Figure 1C). Detailed morphological analysis and grading revealed the development of a wide range of aneurysms in high-dose AngII, varying from type 1 dilated AAA, through most common type 3 dissected, up to type 5-ruptured (Figure 1D). Notably, upon high-dose AngII infusion, the aneurysms started to form by day 14.

In all mice that developed an aneurysm, the expression of *HMOX1* was significantly induced at day 14 (Figure 1E,F). This increase was, however, transient and attenuated at day 28. *HMOX1* induction was not observed in animals subjected to the same treatment that did not develop aneurysm. The immunofluorescent staining of HO-1 in the aorta confirmed these findings. There was a significant upregulation of HO-1 within the aneurysmal tissue, however in a distinguishable subset of cells (Figure 1G). Further analyses revealed that increased HO-1 signal was present in CD45^+^ cells, mostly in CD163^+^ anti-inflammatory macrophages, rather than CD86^+^ proinflammatory macrophages (Figure 1H).

### 3.2. Lack of Ho-1 Protects Against Induction of AAA in Non-Hyperlipidemic Mice; However, If the Aneurysms Form, They Are Unstable and Prone to Rupture

Given a significant upregulation of HO-1 in aneurysms, we decided to use the dose of 2500 ng/kg/min to elucidate further the involvement of HO-1 in AAA in non-hyperlipidemic mice. At this dose, up to day 28, the HO-1 knockout mice tended (*p* = 0.08) to develop AAAs less frequently in comparison with wild-type animals (Figure 2A). However, if formed and unruptured, the aneurysms in HO-1 KO mice were milder, smaller and occurred preferably in the mesenteric artery. Though, in one HO-1-deficient animal, we observed also a severe grade 4 aneurysm, which developed by day 14 (Figure 2B,C). Time course assessment of maximal aortic diameter using ultrasonography (USG), revealed a significant increase in the aortic diameter in WT mice, but not in KO animals, already at day 14 (Figure 2D). In the latter group, in the formed aneurysms, we did not observe a well-defined thrombus (Figure 2E).

Considering a hampered increase in the aortic diameter in HO-1 KO mice, we compared the incidence of aneurysms up to day 14, basing on the USG data. We found a significant decrease in AAA appearance upon HO-1 deletion (Figure 2F). Ex vivo outer aortic diameter measurement indicated a significant increase upon AngII treatment in WT mice, which was abolished by HO-1 deficiency (Figure 2G). However, strikingly, if aneurysms did develop, they were unstable and ruptured (Figure 2H).

### 3.3. Enhanced Immune Cell Infiltration and Oxidative Stress Are Present Only in the Formed Aneurysms

In previous studies, HO-1 deficiency exacerbated AAA formation in AngII-treated ApoE KO mice and porcine elastase model, which was associated with enhanced inflammatory response, macrophage infiltration, and oxidative stress [11,12]. Our data obtained in non-hyperlipidemic mice infused with AngII shows the contrary. Thus, we inspected the same parameters to recognize the basis of the discrepancy.

The analysis of the count and profile of circulating leukocytes on day 14 revealed no significant differences in response to AngII between the wild type and HO-1 deficient mice (Figure 3A). Immune CD45^+^ cells were absent from the aortic wall of animals, which did not develop the aneurysm. On the other hand, in a formed aneurysm, there was a massive infiltration to the aneurysmal tissue, regardless of the genotypes. The immune cells were present at the edge of aneurysmal tissue, and the distance from the primary aortic wall was greater in the KO animal (Figure 3B). Enhanced immune cell infiltration in the aneurysmal tissue was concomitant with a remarkable increase in the expression of inflammatory molecules VCAM-1 and E-selectin (Figure 3C). Similarly, in WT animals, an amplified expression of proinflammatory cytokines IL-1β and IL-6 was visible within the lesions upon aneurysm development. Such a clear correlation was not depicted for IL-4 (Figure 3D).

Further, we assessed the oxidative status of the aortas, using the fluorescent probe detecting reduced thiols (Figure 4). Whereas there was no difference in the redox status when the aorta remained unaltered, aneurysm formation resulted in a significant deterioration in oxidative balance, even more pronounced upon HO-1 deficiency, in accordance with previous data [12].

To sum up, appreciable changes in the inflammatory response and oxidative status are present only in the formed aneurysms. They, therefore, do not seem to account for different susceptibility to AAA. That is why, taking the next steps, we aimed to explore other potential causes of resistance in HO-1-deficient animals to AAA formation.

### 3.4. Ho-1 Deficiency Delays Angii-Induced Blood Pressure Rise, With Concomitant Changes in at2r Expression

Angiotensin II administration elevates systemic blood pressure, which represents one of the main risk factors for AAA development [1]. Therefore, we checked the time-course of blood pressure increase upon AngII infusion. Whereas AngII caused an abrupt blood pressure elevation in WT animals, in HO-1 KO mice, the increase was gradual until day 14, and then the blood pressure remained stable (Figure 5A). Then, we examined the expression of angiotensin II receptors, *AT1R* and *AT2R*, which exert opposite functions, but both are responsible for the AngII-induced regulation of vascular tension and blood pressure [19]. Unlike *AT1R* (Figure 5B), *AT2R* was significantly upregulated in HO-1 KO upon AngII (Figure 5C), which could account for the attenuation of blood pressure rise in HO-1-deficient animals.

### 3.5. AngII Infusion Increases Total Heme Content Especially in AAA

Whereas the cytoprotective effects of HO-1 in AAA are mostly attributed to its products (carbon monoxide and biliverdin—reviewed in [3]), HO-1 is above all a heme-degrading enzyme. Heme has a bimodal function in the cells, varying from devastating oxidative stress induction [20] to protective maintenance of proper mitochondrial metabolism [21]. Hence, we inspected the total heme content and the expression of the heme-related enzymes in the abdominal aortic wall upon AngII infusion.

The obtained results showed that AngII significantly increases heme content in the aortic wall (*p* = 0.03). Moreover, total heme content was 3–5 times higher in mice that developed the aneurysm, irrespectively of genotype (Figure 6A). AngII increased the total heme level in WT animals regardless of AAA formation (*p* < 0.05 for non-AAA, *n* = 3; *p* < 0.01 for both non-AAA+AAA, *n* = 6). On the contrary, aortic heme content was higher (*p* = 0.028) in saline-infused HO-1-deficient animals but was not significantly increased upon AngII infusion (*p* = 0.3). The analysis of the expression of *ALAS1*, *FLCVR1*, and *SLC40A1* (ferroportin) revealed no significant differences (Figure 6B–D). The gene expression pattern in WT mice that developed the aneurysm suggests a downregulation of heme synthesis with a concomitant increase in both heme and iron export. Similarly, reduction of the heme synthesis rate-limiting gene (*ALAS1*) is visible in the saline-treated HO-1 KO mice.

### 3.6. AngII Infusion Leads to the Thickening of Aortic Media in Ho-1 Ko Mice

Hematoxylin and eosin staining revealed significant changes in the aortic morphology, especially in the media layer. AngII promoted thickening of the media only in HO-1 KO animals, with a concomitant perpendicular orientation of smooth muscle cells towards elastin fibers (Figure 7A,B). We assume it might be caused by a higher proliferation of smooth muscle cells [22]. Indeed, in vitro studies showed that AngII promoted the division of human aortic smooth muscles and the effect was much more pronounced in HO-1-deficient cells (Figure 7C). In aortas, however, despite an overall increased proliferation in response to AngII, there were no significant differences between genotypes in the expression of proliferation marker Ki-67 (Figure 7D).

### 3.7. Differential Effect on the Collagens Upon AngIIInfusion between the Genotypes

Augmented collagen turnover grounds the pathogenesis of aneurysm [23]. Moreover, mechanical properties and stability of vessels strongly rely on collagen structure [24]. Therefore, to analyze the differential effects of HO-1 deficiency on AAA formation and rupture, we inspected the collagen levels.

Trichrome staining unveiled that AngII increased total collagen content mainly in WT animals (Figure 8A). Interestingly, in WT mice, AngII infusion induced mostly the expression of collagen I (Figure 8B,C), while in their HO-1 KO counterparts, the upregulated isoform was collagen III (Figure 8D). The immunofluorescent showed a similar pattern (Figure 8E). 

Then, we investigated the level of collagen regulators. The expression of *TGFB1* and *TGFB2* did not differ significantly between the groups. Albeit, in animals that developed an aneurysm, they were upregulated (Figure 9A,B). In HO-1 KO mice, AngII strongly induced PAI expression, and overall, the animals with aneurysm had a lower level of *PAI* (plasminogen activator inhibitor, Figure 9C). On the other hand, u-PA (urokinase-type plasminogen activator; encoded by *PLAU* gene) tended to rise in WT animals upon AngII infusion.

Finally, AngII induced expression of osteopontin (encoded by *SPP1* gene) was increased both at mRNA and protein level. The induction was statistically significant in both genotypes, but much more pronounced in WT animals (Appendix A). Similarly, AngII promoted the expression of osteopontin receptor—CD44, but only in WT animals (Appendix A).

### 3.8. MMP2 and TIMP2 are Strongly Upregulated Upon AngII in HO-1-Deficient Mice

Collagen content and AngII-induced aortic rearrangements are closely related to the balance between the activity of metalloproteinases (MMPs) and their inhibitors (TIMPs) [25]. We found that AngII promoted the expression of metalloproteinases: *MMP2* in HO-1 KO, and *MMP3* in WT animals. No significant differences were observed in *MMP9* levels (Figure 10A), known to be highly expressed in aortas [25].

Further, we performed in situ zymography (ISZ) and co-stained samples for MMP2 and MMP9. AngII enhanced gelatinases activity. Whereas. in formed AAA. it spread almost through the whole aneurysmal tissue; in the unexpanded aortas, it was rather site restricted. The signal for MMP2, but not MMP9 colocalized with gelatinases activity in HO-1-deficient animals. In WT mice, there were small areas of colocalization only with MMP2 (Figure 10B). The expression of TIMPs was also dependent on genotype: induction of *TIMP1* by AngII was stronger in WT than in HO-1 KO animals. In saline-treated KO mice, *TIMP2* tended to be higher in comparison to WT counterparts. The difference was significantly heightened upon AngII infusion. The level of *TIMP3* remained unaltered (Figure 10C).

## 4. Discussion

The infusion of AngII to normolipidemic C57Bl6 mice as a model of AAA was introduced around 20 years ago [13,26]; yet, it is not commonly used due to lower incidence of AAA in comparison to hyperlipidemic mice [27]. Nevertheless, the pathological changes, including elastin degradation, immune cell infiltration, or expansion of abdominal aortic wall, do not significantly differ between those two models. In this study, we demonstrate the similar incidence of AAA in mice on mixed genetic background C57Bl6xFVB in comparison to C57Bl6, which could be further augmented by increasing the dose of AngII. The use of such mixed background permits efficient breeding of HO-1-deficient mice, and it has a negligible effect on the basal cytokine level in comparison to C57Bl6 mice [28]. AngII-based models can also be considered as models of aortic dissection [29] due to formation of medial tears, through which the blood penetrates and subsequently traps the blood components within the aortic wall. Heme released from damaged erythrocytes may then lead to activation of HO-1.

The impact of HO-1 deficiency has already been studied in two murine models, characterized by high AAA incidence: porcine elastase infusion (AAA forms in up to 100% males) and AngII infusion to hyperlipidemic ApoE KO mice (80% formation of AAA) [11,12,27]. Both models suggest that lack of HO-1 exacerbates AAA formation. In accordance, the relevance of HO-1 in AAA biology was addressed using HO-1 modulators, and the HO-1 upregulation was concomitant with a positive impact on AAA [reviewed in [30,31].

In our study, we decided to use non-hyperlipidemic animals to infer about the disease susceptibility. We show a dual role of HO-1 deficiency, which, on the one hand, protects against AAA development, but still, on the other, may aggravate the already formed aneurysms and promote their rupture. Interestingly, such seemingly irreconcilable results between mice strains have been previously reported in the case of the use of anti-TGFβ antibodies [15,32]. Further understanding of the reason behind these discrepancies may pave the way to determine AAA pathophysiology.

In the previous reports, HO-1 deficiency-induced exacerbation of AAA formation was associated with enhanced inflammatory response, macrophage infiltration, and oxidative stress [11,12]. Thus, we decided to inspect those parameters. Basing on our previous experience [18], and bearing in mind that aneurysms are formed by day 14, we focused on day 14 after implementation of osmotic pumps. We found that enhanced inflammatory response and oxidative imbalance are distinctive features of an aneurysm itself. In line with the previous report [12], in our hands, HO-1 deficiency led to intensified oxidative stress and altered distribution of immune cells, which could aggravate the state of the aortas. Such changes might also compromise the aortic integrity, and thus explain the susceptibility to aneurysm rupture.

Undoubtedly, infiltration of immune cells to the aortic wall is the prerequisite of the remodeling and loss of medial smooth muscle cells [33,34]. Our data show that the increase in the level of *HMOX1* results mainly from the infiltration of immune cells to the aortic wall and not from its upregulation in regular aortic cellular components. The previous data were obtained in the models, in which the infiltration of immune cells is facilitated. In the elastase mode, there is an artificial imposition of the aortic rearrangement and the induction of vascular damage. Such a procedure may accelerate immune cell homing to the aortic wall and aortic rearrangement, and thus impede the verification of the possible mechanisms underlying altered susceptibility to AAA formation. Accordingly, the ApoE KO mice have much more immune cells, including macrophages, in the aorta in comparison to WT mice [35,36], which could immediately respond to AngII and induce the remodeling.

Ang II concentration in plasma in physiological state is around 50 fmol/mL and rises to 100 fmol/mL in hypertension. The kidney levels are approximately 2–3 times higher [37]. The commonly used doses in biomedical research greatly exceed those values; however, interestingly AngII-induced hypertension is characterized by an only 2-fold increase in AngII plasma content (reviewed in [38]). Additionally, the application of angiotensin-converting enzyme inhibitors abrogates the induction of hypertension, implying that endogenous AngII generation is indispensable for the process [39]. Irrespectively of the dose (varying between 200–2500 ng/kg/min), AngII infusion leads to around 40% increase in systolic blood pressure [18,35,36]. It substantiates that AAA formation is not solely dependent on hemodynamic parameters. The cellular effects and impact on angiotensin receptors shall have the most prominent effect on observed changes in hypertension or AAA formation. On the other hand, mitigated blood pressure elevation in HO-1-deficient mice may be beneficial as it gives time for the vessel to adapt and possibly to decrease the vascular injury and remodeling [40]. Our results suggest that the delayed blood pressure response to AngII in HO-1-deficient mice might be caused by changed angiotensin receptor profile, especially the enhanced *AT2R* expression, which inversely correlates with blood pressure values [38]. Finally, the observed difference in SBP substantiates the impact of HO-1 in blood pressure regulation.

Intraperitoneal injections of heme were shown to decrease AAA development, and the protective effect was attributed to HO-1 induction [11]. Here, we employed fluorescence-based measurement of heme, which pointed to a general upregulation of heme content upon AngII infusion, especially in AAA. However, this method, despite high sensitivity, is less specific and does not differentiate between heme and hemoproteins [41]. It would be valuable to inspect the level of labile heme in situ to elucidate its impact on AAA, further focusing on several aspects such as HO-1 activity. Interestingly, in WT animals which developed the aneurysm, we observed the pattern of expression of heme-related enzymes which indicated an overall downregulation of heme synthesis, increased heme and iron export. It could indicate for proper sensing of augmented heme content and substantiates the induction of HO-1 enzymatic activity during AAA formation. In HO-1 KO mice expression of ALAS1, the rate-limiting enzyme in heme synthesis was very low already in steady-state conditions, which stays in accordance with increased heme content in this group of mice. At the examined timepoint, we observe a downregulation of heme synthesis, however, given significantly higher heme content in upon AngII, it is highly plausible that during the earlier timepoints, the impact on heme synthesis was the opposite.

Literature data clearly state that the mechanical resistance of the vessel wall is highly reliant on the collagen network. The progression of AAA and its susceptibility to rupture is contingent on fine-tuned collagen turnover. In the aortic wall, the load-bearing proteins are proteolysis-resistant fibrillar collagens type I and III [23,42]. Here, we show that while in WT animals, AngII induced mostly the expression of collagen I, in HO-1 KO mice it led to elevated collagen type III expression. The latter could account for increased predisposition to rupture, as it is found in rupture-prone sites [43,44]. In WT mice, the AngII-dependent induction of collagen may be dependent on osteopontin expression [45], rather than TGFβ signaling. Importantly, TGFβ neutralization augments AAA incidence [15].

During the study, we noticed a significant AngII-induced thickening of media upon AngII in HO-1 KO mice. It was mostly caused by the different orientation of SMCs. Instead of being parallel, SMCs were perpendicular to the lumen. In all the animals, they were localized between the 2nd and 3rd elastin layer; however, in some mice, they were extended throughout the whole vessel wall. Such disarray has already been observed in thoracic aneurysm patients with *MYH11* and *ACTA2* mutations [22].

The aortic SMCs in AngII-infused HO-1 KO mice were not in close contact with each other; instead, there were separated by acellular spaces. Presence of acellular areas in aneurysms is well known as idiopathic medial cystic degeneration or mucoid ECM accumulation [46]. In general, the existence of pores in the wall influences wall deformation characteristics, which may increase the risk of strain-induced rupture [47]. Interestingly, the low dose of AngII (200 ng/kg/min) leads to media thickening in rats with a concomitant increase in MMP2 and *TIMP2* level [48].

MMP2 and MMP9 are known to be pivotal in human aneurysm development [49]. In this study, we did not detect any differences in MMP9, but we observed a significantly higher level of MMP2 in AAA-resistant HO-1 KO mice. Moreover, in situ zymography revealed that MMP2 is involved in the aortic wall remodeling. In our previous study, we observed colocalization of gelatinase activity with MMP9 signal in mice without aneurysms [18]. Both MMPs act on predigested, solubilized monomers of collagen [50,51,52]. Hypothetically, the shortage of collagenases or other metalloproteinases could account for the lack of aortic expansion in HO-1-deficient mice. Conversely, given the relatively straightforward correlation of aneurysm incidence with *MMP3* and *TIMP1* level, we suppose that these may be the key players in the aortic remodeling upon high-dose AngII. Notably, MMP3 activity is essential for the development of AngII-induced aneurysms [33], and it is strongly reduced in AngII-treated HO-1 KO mice.

Undeniably, the infusion of AngII in HO-1-deficient animals induced some aneurysmal features, such as the presence of acellular spaces, increased MMP2 expression, and altered collagen content. Still, despite those characteristics, the aortic diameter remained unchanged. It may indicate for a presence of a ‘switch’ which keeps the aorta restrained, preventing its expansion. In our study, we observed a differential and prominent induction of *PAI*. It stays in line with previous data reporting an inverse association between HO-1 expression and PAI induction in the injured aortic walls [53]. Interestingly, the overexpression of PAI protects against AAA [54]. Thus, it may be beneficial to elucidate its impact in HO-1-deficient aortas further. Another candidate for a ‘switch’ might be TIMP2. Levels of both PAI and TIMP2 are lower in aneurysmal in comparison with athero-occlusive abdominal aortas [55]. Furthermore, female aortas have significantly higher expression of those genes in comparison to males [56,57]. Importantly, similarly to HO-1 deficient mice, females are less prone to AAA development; however, if formed, it is more vulnerable and susceptible to rupture [1]. Apart from the presence of such a single switch, there may be a more complex compensatory mechanism, which could help in the maintenance of aortic homeostasis. One of them would be increased aortic stiffness, which could counteract intensified aortic remodeling [58].

### Limitations of the Study

This research has several limitations. To increase AAA incidence, we used a high dose of AngII, which had not been regularly used in previous studies. It would be essential to perform additional experiments to define this model and outline any possible unspecific side-effects. Secondly, we used mice having the global knockout of HO-1. Identification of exact cell population, which is responsible for the observed phenomenon, would be of utmost importance. Especially, the role of CD163^+^ macrophages, which infiltrate aneurysmal tissue and highly express HO-1, and which can disappear in HO-1 KO mice [59,60], should be considered. It could possibly account for the dual impact of HO-1 deficiency in this investigation setup. Moreover, we focused on day 14 and 28, but earlier time-points could give further insights into the mechanisms of AAA-induction, while a prolongation of the study could provide some additional information about lesion stability. Finally, we aimed to provide some mechanistic explanations for the observed phenomenon, yet our conclusions are based on correlations and accompanying effects. Thus, their experimental verification should be the next step.

## 5. Conclusions

To conclude, high-dose of angiotensin II leads to efficient formation of AAA in non-hypercholesterolemic mice, which is attenuated by HO-1 deficiency. Yet, if AAA is formed, it is significantly more prone to rupture upon HO-1 shortage. Differential susceptibility to AAA formation is not dependent on inflammatory response or oxidative stress induction. AAA-resistant mice are also characterized by enhanced *PAI*, *TIMP2*, and *AT2R* expression, significant medial thickening, and delayed blood pressure elevation in response to AngII. The evidenced dual role of HO-1 may result from multiplicitous events associated with AAA formation. Whereas, during the initial steps, HO-1 abundant macrophages trigger the remodeling of the aortic wall, during the further steps, they may be also responsible for the proper disposal of harmful molecules ensuring AAA stability. Undoubtedly, the role of HO-1 is equivocal and shall be addressed in more detail during every step of AAA initiation and stability.

## Figures and Tables

**Figure 1 cells-10-00163-f001:**
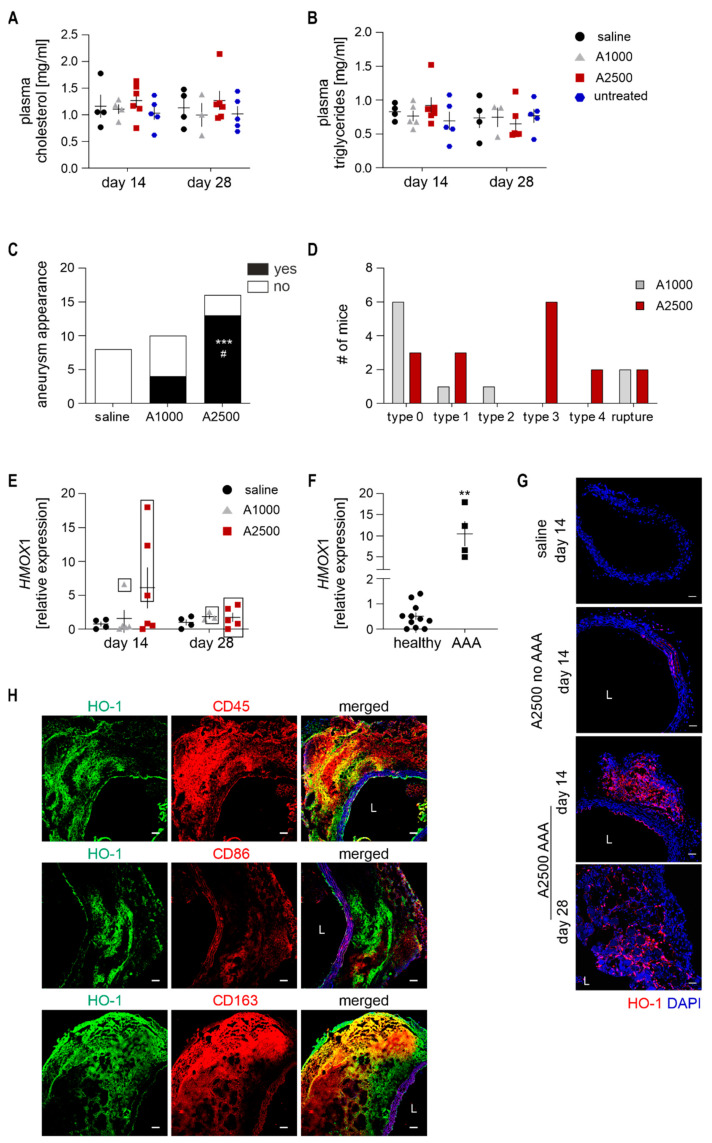
Heme oxygenase 1 is upregulated during the angiotensin II-induced formation of abdominal aortic aneurysm (AAA) in non-hypercholesterolemic mice. The mice were infused with either saline or angiotensin II at the dose 1000 or 2500 ng/kg/min using the osmotic pumps for up to 28 days, as shown in Appendix A. (**A**) Plasma cholesterol level assessment in comparison to untreated animals; three-way ANOVA+Tukey’s; (**B**) Plasma triglycerides level assessment in relation with untreated animals; three-way ANOVA+Tukey’s post hoc; (**C**) The frequency of aortic aneurysm appearance. Fisher’s exact test, *** *p* < 0.001 vs. saline; ^#^
*p* < 0.05 vs. A1000; (**D**) Grading of developed aneurysms. The detailed criteria are presented in the methods section; (**E**) Relative expression of * HMOX-1 * in the abdominal aortic wall at day 14 and 28 after initiation of AngII infusion. *eEF2* was used as a reference gene. Two-way ANOVA with Tukey’s post hoc test. Rectangle—mice, which developed an aneurysm; (**F**) Comparison of the relative expression of * HMOX-1 * in the abdominal aortic wall at day 14 in mice with and without AAA. *eEF2* was used as a reference gene. Mann–Whitney test, ** *p* < 0.01 AAA vs. no AAA; (**G**) Immunofluorescent staining of HO-1 in the aorta. Red—HO-1; blue—DAPI; L—lumen. Representative images; scale bar 50 µm; (**H**) Immunofluorescent co-staining of HO-1 and immune cell markers (CD45, CD86, CD163) in the aneurysmal tissue. Green—HO-1, red-CD45/CD86/CD163, blue—DAPI; L—lumen. Representative images, scale bar 50 µm.

**Figure 2 cells-10-00163-f002:**
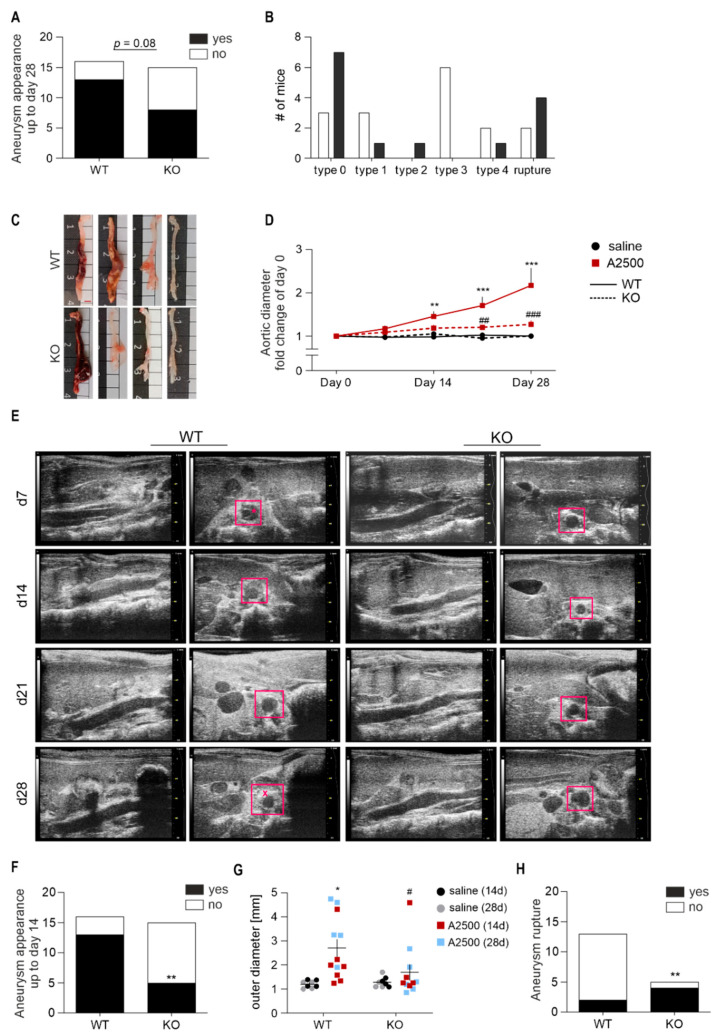
Lack of HO-1 protects against induction of AAA in non-hyperlipidemic mice; however, if the aneurysms form, they are unstable and prone to rupture. The mice were infused with either saline or angiotensin II at the dose 2500 ng/kg/min using the osmotic pumps for up to 28 days. (**A**) The frequency of aortic aneurysm appearance, Fisher’s exact test; (**B**) Grading of developed aneurysms. The detailed criteria are presented in the methods section; (**C**) representative image of aortas isolated from WT and HO-1 KO mice infused with AngII. Scale bar = 2.5 mm; (**D**) Time-dependent changes in the aortic inner diameter measured with USG. Two-way ANOVA+ Tukey’s post hoc test, ** *p <* 0.01; *** *p <* 0.001 vs. saline; *^##^ p <* 0.01; ^###^
*p <* 0.001 vs. WT. (**E**) Representative images of aneurysm formation in WT and HO-1 KO mice; rectangle—aorta; *—thrombus; x—stabilized thrombus; (**F**) The frequency of aortic aneurysm appearance up to day 14, basing on USG results, Fisher’s exact test, ** *p* < 0.01 vs. WT; (**G**) Outer aortic diameter of aortas collected and measured ex vivo at day 14 and 28 of the study. Two-way ANOVA with Tukey’s post hoc test, * *p* < 0.05 vs. saline; ^#^
*p* < 0.05 vs. WT; (**H**) The frequency of aortic aneurysm rupture, Fisher’s exact test, ** *p* < 0.01 vs. WT.

**Figure 3 cells-10-00163-f003:**
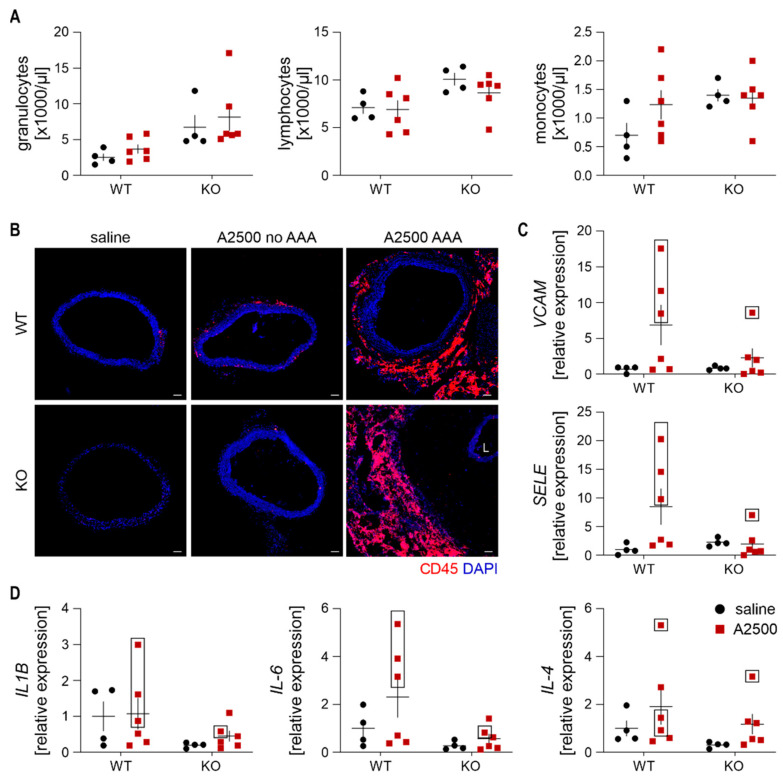
Enhanced immune cell infiltration is present only in the formed aneurysms. The mice were infused with either saline or angiotensin II at the dose 2500 ng/kg/min using the osmotic pumps for 14 days. (**A**) Quantification of circulating white blood cells subpopulations. Two-way ANOVA with Tukey’s post hoc test. (**B**) Immunofluorescent staining of infiltrating CD45^+^ cells within the aortic wall. CD45—red, nuclei- blue; L—lumen, magnification 200×. Scale bar 50 µm. (**C**) Relative expression of *VCAM1* and *SELE* in the abdominal aortic wall. *eEF2* was used as a reference gene. Two-way ANOVA with Tukey’s post hoc test; (**D**) Relative expression of *IL-1b*, *IL-4*, and *IL-6* in the abdominal aortic wall. *eEF2* was used as a reference gene. Two-way ANOVA with Tukey’s post hoc test. Rectangle—mice which developed the aneurysm.

**Figure 4 cells-10-00163-f004:**
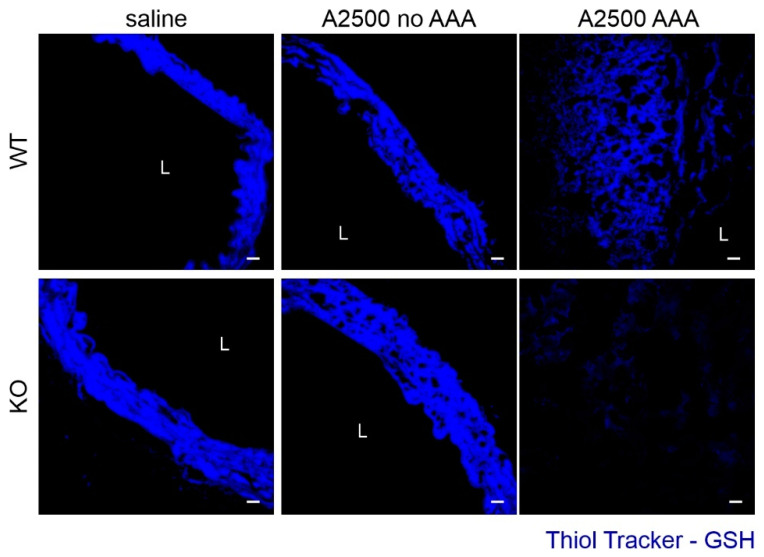
Enhanced oxidative stress is present only in the formed aneurysms. The mice were infused with either saline or angiotensin II at the dose 2500 ng/kg/min using the osmotic pumps for 14 days. Redox status of the aortas was assessed using Thiol Tracker, which detects reduced thiols. Representative pictures, scale bar 50 µm.

**Figure 5 cells-10-00163-f005:**
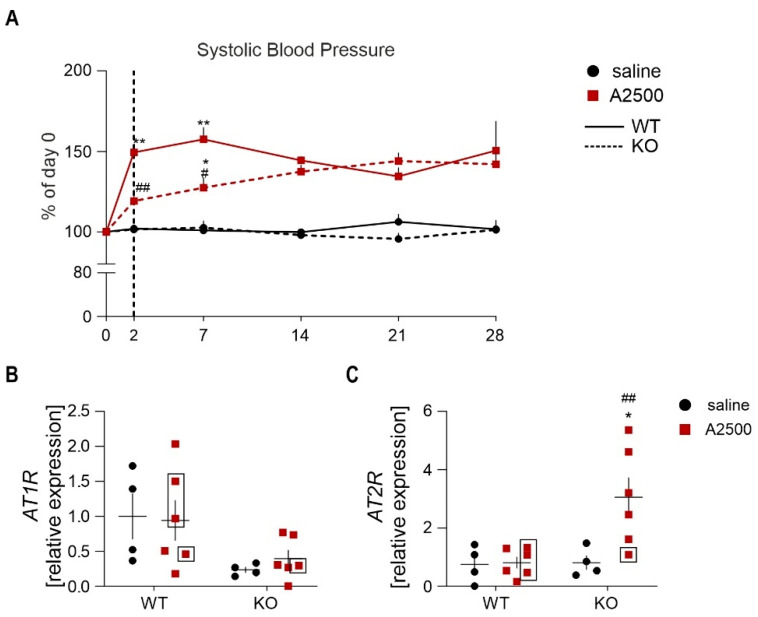
HO-1 deficiency delays AngII-induced abrupt blood pressure rise with concomitant changes at AT2R expression. (**A**) Time course of changes in systolic blood pressure (SBP) in WT and HO-1 KO, presented as a change of day 0; * *p* < 0.05 vs. saline; ** *p* < 0.01 vs. saline; ^#^
*p* < 0.05 vs. WT; ^##^
*p* < 0.01 vs. WT. (**B**,**C**) Relative expression of (**B**) *AT1R* and (**C**) *AT2R* in the abdominal aortic wall at day 14 after placement of osmotic pumps. *eEF2* was used as a reference gene. Two-way ANOVA with Tukey’s post hoc test. Rectangle—mice which developed the aneurysm. * *p* < 0.05 vs. saline; ^##^
*p* < 0.01 vs. WT.

**Figure 6 cells-10-00163-f006:**
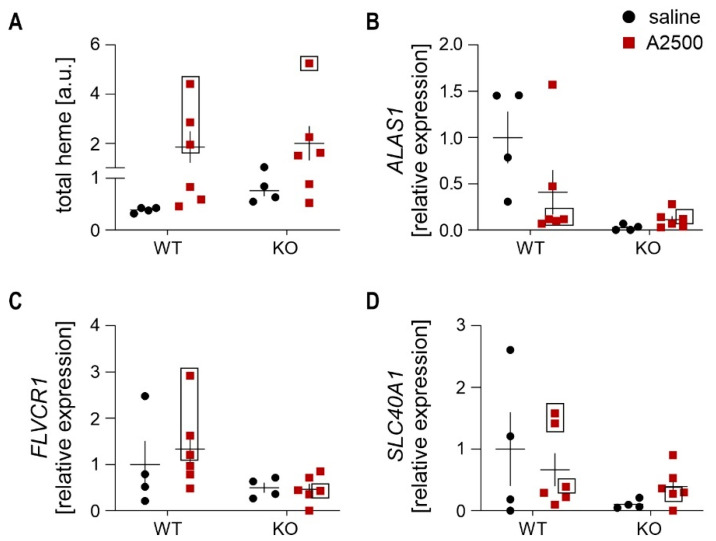
AngII infusion increases total heme content especially in AAA. The mice were infused with either saline or angiotensin II at the dose 2500 ng/kg/min using the osmotic pumps for 14 days. (**A**) Total heme content normalized to protein level. Two-way ANOVA with Tukey’s post hoc test. (**B**–**D**) Relative expression of (**B**) *ALAS1*; (**C**) *FLVCR1*; and (**D**) *SLC40A1* in the abdominal aortic wall at day 14 after placement of osmotic pumps. *eEF2* was used as a reference gene. Two-way ANOVA with Tukey’s post hoc test. Rectangle—mice which developed the aneurysm.

**Figure 7 cells-10-00163-f007:**
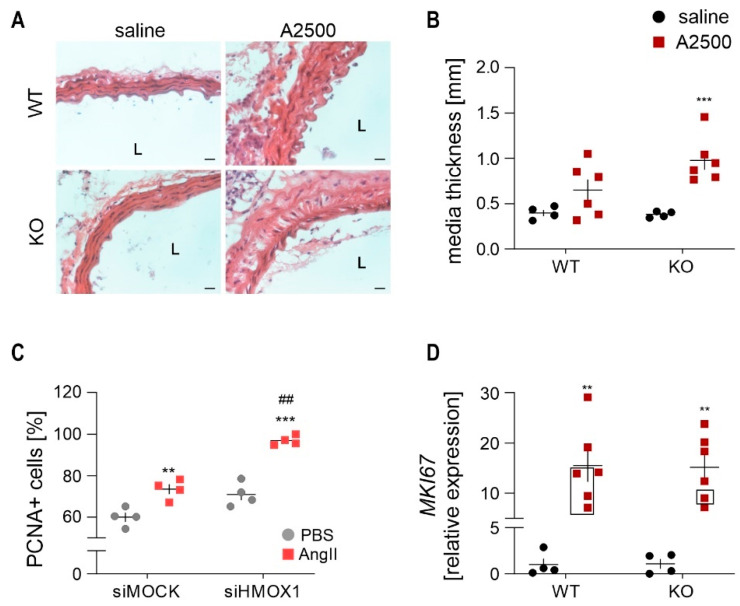
AngII infusion to HO-1 knockout mice triggers significant thickening of media. The mice were infused with either saline or angiotensin II at the dose 2500 ng/kg/min using the osmotic pumps for 14 days. (**A**) Haematoxylin and eosin staining of aortas. Representative images; magnification 400×; scale bar = 50 µm; (**B**) Quantification of media layer thickness of abdominal aortas; Two-way ANOVA with Tukey’s post hoc test. *** *p* < 0.001 vs. saline; (**C**) Percentage of proliferating human aortic smooth muscle cells in response to 24 h stimulation with 50 nM AngII. Number of proliferating cells was assessed by PCNA—proliferating cells nuclear antigen staining; Two-way ANOVA with Tukey’s post hoc test. *** *p* < 0.001 vs. PBS, ^##^
*p* < 0.01 vs. siMock; (**D**) Relative expression of proliferation marker *MKI67* in the abdominal aortic wall. *eEF2* was used as a reference gene. Two-way ANOVA with Tukey’s post hoc test, ** *p* < 0.05 vs. saline; rectangle—mice which developed the aneurysm.

**Figure 8 cells-10-00163-f008:**
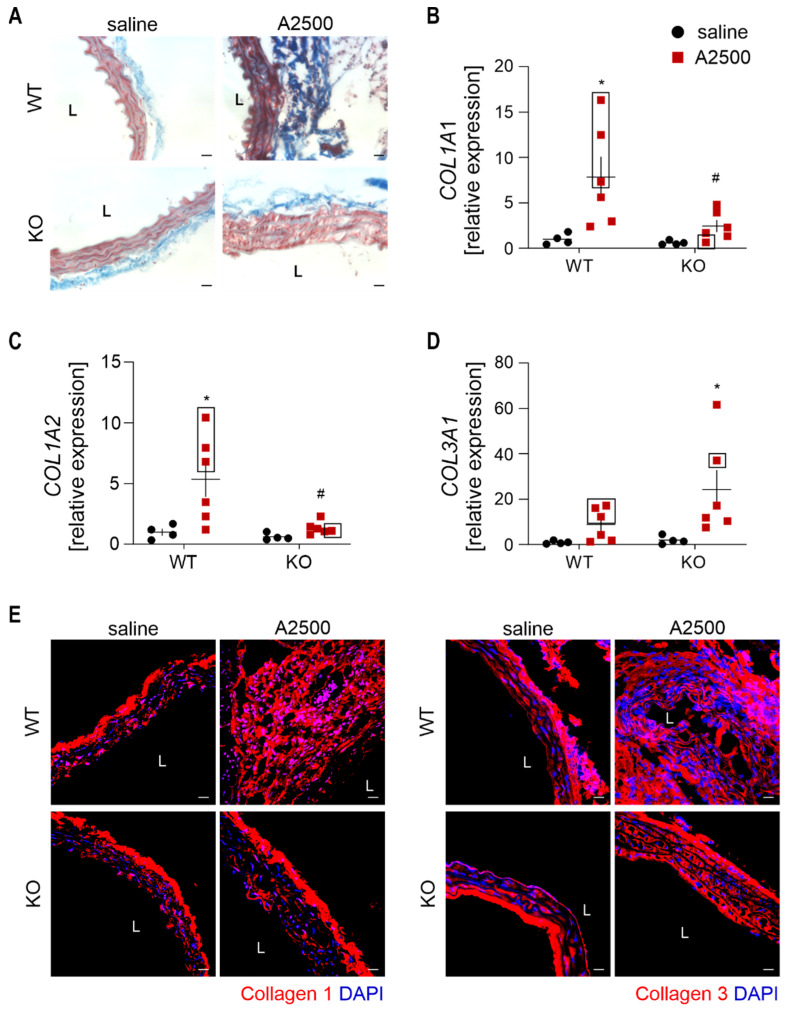
Differential effect on the collagens upon AngII infusion between the genotypes. The mice were infused with either saline or angiotensin II at the dose 2500 ng/kg/min using the osmotic pumps for 14 days. (**A**) Trichrome staining in the abdominal aorta; Representative images, magnification 400x; (**B**–**D**) Relative expression of (**B**) *COL1A1*; (**C**) *COL1A2*; and (**D**) *COL3A1* in the abdominal aortic wall. *eEF2* was used as a reference gene. Two-way ANOVA with Tukey’s post hoc test; * *p* < 0.05 vs. saline, ^#^
*p* < 0.05 vs. WT. Rectangle—mice which developed the aneurysm; (**E**) Immunofluorescent staining of collagen I and collagen III within the abdominal aorta. L—lumen. Collagen—red, nuclei—blue. Representative images. Scale bar 30 µm.

**Figure 9 cells-10-00163-f009:**
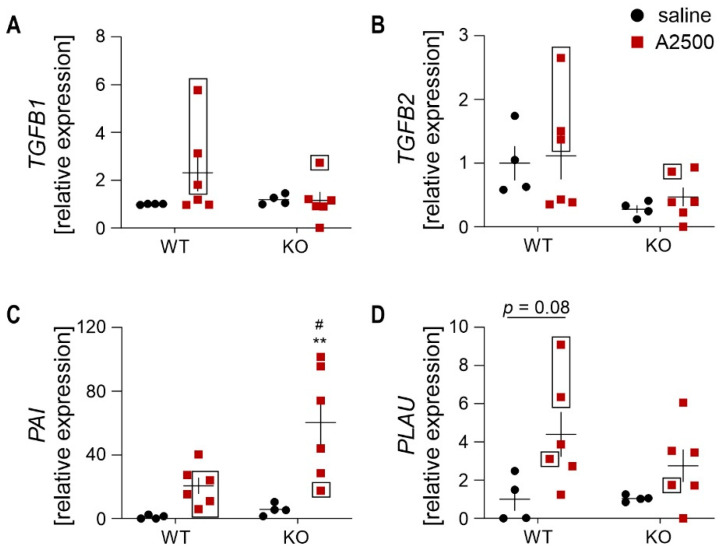
TGFβ-dependent signaling upon AngII. (**A****–D**) Relative expression of (**A**) *TGFB1* (**B**) *TGFB2*; (**C**) *PAI1*; and (**D**) *PLAU* in the abdominal aortic wall at day 14 after placement of osmotic pumps. *eEF2* was used as a reference gene. Two-way ANOVA with Tukey’s post hoc test; ** *p* < 0.01 vs. saline, ^#^
*p* < 0.05 vs. WT. Rectangle—mice which developed the aneurysm.

**Figure 10 cells-10-00163-f010:**
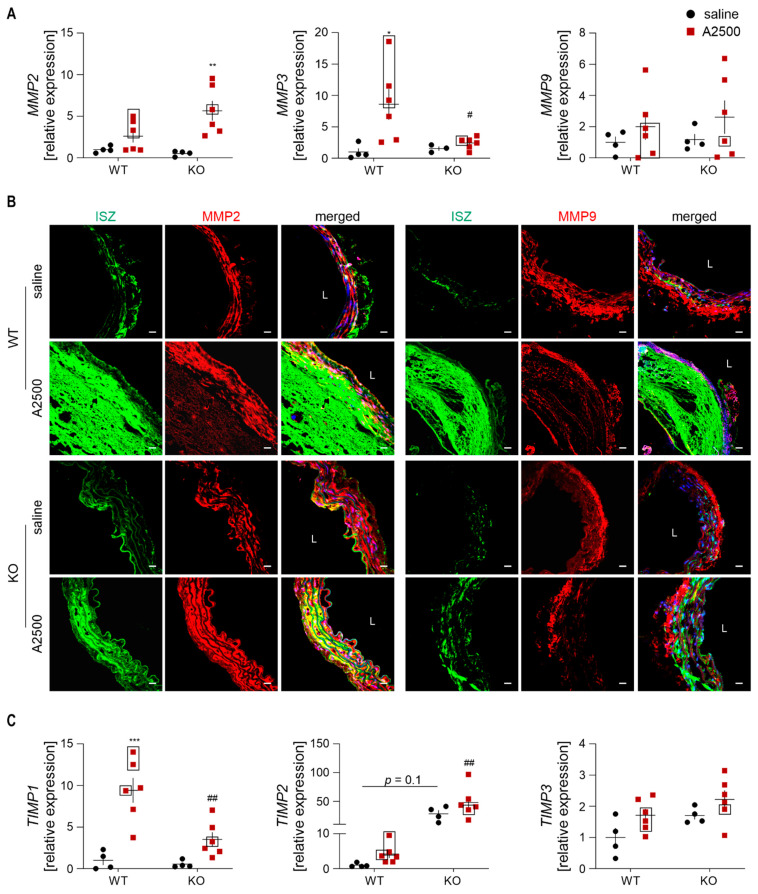
MMP2 and TIMP2 are strongly upregulated upon AngII in HO-1-deficient mice. (**A**) Relative expression of *MMP2*, *MMP3*, and *MMP9* in the abdominal aortic wall. *eEF2* was used as a reference gene. Two-way ANOVA with Tukey’s post hoc test; * *p* < 0.05 vs. saline, ** *p* < 0.01 vs. saline, ^#^
*p* < 0.05 vs. WT. Rectangle—mice which developed the aneurysm; (**B**) The activity of gelatinases (green) and MMP-2/MMP-9 level (red). Blue—nuclei. The activity of gelatinases was assessed by in situ zymography. Representative images. Scale bar = 30 µm; L—lumen; (**C**) Relative expression of *TIMP1*, *TIMP2*, and *TIMP3* in the abdominal aortic wall. *eEF2* was used as a reference gene. Two-way ANOVA with Tukey’s post hoc test; *** *p* < 0.001 vs. saline, ^##^
*p* < 0.01 vs. WT.; Rectangle—mice which developed the aneurysm.

## Data Availability

The data used to support the findings of this study are available from the corresponding author upon request.

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
