# Peer review of "A Dual Role of Heme Oxygenase-1 in Angiotensin II-Induced Abdominal Aortic Aneurysm in the Normolipidemic Mice"

_cells, 2021, doi:10.3390/cells10010163_

Round 1

Reviewer 1 Report

The present paper deals with the dual role of heme oxygenase-1 in angiotensin II-induced abdominal aortic aneurysm in the normolipidemic mice. Interesting findings to the field are described and shed light on the dual role of HO-1 in AAA so that the manuscript is convincing and worthy of publication. Nevertheless, in my opinion, at least one major modification is needed to improve the quality of the paper.

Since they used either wild-type mice or mice having the global knockout of HO-1, it would be of great interest the use of wild-type mice treated with a specific HO-1 inhibitor thus creating a transient HO-1 inhibition and allowing a more realistic interpretation of the results.

In addition minor modification can improve the quality of the manuscript.

In the abstract the following sentence is not very clear please rephrase "We demonstrated that the high-dose of AngII leads to the efficient formation of AAA in non-hypercholesterolemic mice, which is attenuated by HO-1 deficiency."

In the abstract and all along the text please avoid to use acronym without explaining their meaning at least the first time they appear. As an example in the abstract "increased PAI, TIMP2 and AT2R expression"

In the introduction, line 48 please add more detailed description of HO-1 and its physiological role beside AAA, implications for HO-1 inhibition by small molecules and in knock-out animal model , activity of metabolites etc.

Please check the manuscript for small typos.

Reviewer 2 Report

The study made by Kopacz et al., investigate the role of heme oxygenase (HO-1) in AAA. To develop their hypothesis the authors used normolipidemic mice. The authors concluded that HO-1 has dual role; it can protect against AAA development but exacerbates the state of AAA once formed. The idea of the study is very appealing; however, the authors need to be more careful in the way they present their data. Some data can go as supplement material. The discussion should be more focused.

  1. The animals used in this study come from a mixed background (C57/BL6 and FVB). However, the author did not clarify if HO-1−/− mice and HO-1+/+ littermates have the same mixed background. This is very important specially because mice on C57 background are not a good model to study AAA.
  2. Figure 1G. it is surprising that HO-1, which is expressed in both endothelial cells and smooth muscle cells, was not detected in saline group. Additionally, it is well known that Ang II infusion affect the level of HO-1, independently if there is AAA or not. The image from A2500 no AAA day 14, doesn’t show any HO-1 expression and did not show any difference compared to Saline 14 days. Please elaborate
  3. Figure 1G. is not clear specifically the comparison between day 14 and day 28. Doing immunohistochemistry will show better morphology and it will make it easier to spot the elastic fiber and the AAA degree. If authors can show Figure 1G at higher magnification, like figure 1H, this will make it more convincing for the readers.
  4. Figure 3B is not clear. We don’t see CD45 yellow. Additionally, KO A2500 AAA is not clear. The authors are encouraged to revise all the immunostaining figures.
  5. Same problems occur in Figure 4. Additionally, Figure 4 is over saturated. One question related to this figure is why there was no vessels remodeling after high dose Ang II treatment?
  6. HO-1 is known to play a crucial role in regulating BP. Surprisingly, there was no difference in the SBP between the WT and KO mice treated with high dose of Ang II. The authors did not discuss this point.
  7. The author should explain how the total heme was increased without any change in the level of gens related to heme synthesis.
  8. In figure 7A, we saw vascular remodeling, which is not the case in the other vessel’s images presented in the previous figures. Please elaborate.
  9. Authors should discuss the latest papers related to HO-1 and AAA. this will make the discussion section. 
  10. A schematic figure summarizing the finding is encouraged.

       Minor revision:

  1. In line 191, please change Individuals to mice

Round 2

Reviewer 1 Report

The paper can be accepted in the present form.

Reviewer 2 Report

I don't have additional comments